# Applicability of Multi-Agent Systems and Constrained Reasoning for Sensor-Based Distributed Scenarios: A Systematic Mapping Study on Dynamic DCOPs

**DOI:** 10.3390/s21113807

**Published:** 2021-05-31

**Authors:** Jose Barambones, Ricardo Imbert, Cristian Moral

**Affiliations:** Madrid Human-Computer Laboratory, Universidad Politécnica de Madrid, 28660 Boadilla del Monte, Spain; ricardo.imbert@upm.es (R.I.); cristian.moral@upm.es (C.M.)

**Keywords:** intelligent sensor networks, distributed problem solving, sensors on dynamic environments, systematic mapping study, distributed constraint optimisation problem

## Abstract

*Context*: At present, sensor-based systems are widely used to solve distributed problems in changing environments where sensors are controlled by intelligent agents. On Multi-Agent Systems, agents perceive their environment through such sensors, acting upon that environment through actuators in a continuous cycle. These problems have not always been addressed from an ad-hoc perspective, designed specifically for the circumstances of the problem at hand. Instead, they have been modelled under a common mathematical framework as distributed constrained optimisation problems (DCOP). *Objective*: The question to answer is how sensor-based scenarios have been modelled as DCOPs in changing environments known as Dynamic DCOP and what their trends, gaps, and progression are. *Method*: A systematic mapping study of Dynamic DCOPs has been conducted, considering the scattered literature and the lack of consensus in the terminology. *Results*: Given the high complexity of distributed constraint-based problems, priority is given to obtaining sub-optimal but fast responses with a low communication cost. Other trending aspects are the scalability and guaranteeing the solution over time. *Conclusion*: Despite some lacks in the analysis and experimentation in real-world scenarios, a large set that is applicable to changing sensor-based scenarios is evidenced, along with proposals that allow the integration of off-the-shell constraint-based algorithms.

## 1. Introduction

Sensor-based systems allow for more complex operations when perceptions are obtained from several distributed sensors and these are combined. In intelligent sensor networks, a sensor is connected with its neighbours within a network, and each node is aware not exclusively of its sensing perceptions, but also capable of self-reasoning in regard to its coordination with its teammates [1,2]. In the field of Artificial Intelligence, the sensing and reasoning of multiple such sensors as “aware” nodes to reach common goals is addressed by distributed systems based on *Multi-Agent Systems* (MAS). In a nutshell, MAS comprised a number of agents that require the ability to coordinate and negotiate with each other considering a set of different goals and motivations, aimed to cooperate beyond the capabilities of any individual agent [3,4].

The behaviour of an agent is composed of a continuous sensor–cognition–action cycle, which integrates the sensing process with the reasoning and execution of the action. During the sensor (or perception) process, the agent acquires, abstracts, and filter the sensed data before sharing them with its neighbours. The action process controls the execution of external acts on effectors upon their environment. The cognition process is aimed to interpret such perception, solve the problem, and perform a plan involving perceptions and actions [4,5]. The literature shows how agents and MAS are used in sensor-based systems as intelligent sensor networks, such as Wireless and Mobile Sensor Networks [1,6,7,8,9,10], IoT orchestration [11,12,13], and in-the-field exploration, localisation, and tracking [14,15,16,17,18], inter alia. Indeed, MAS are widely used in environments where certain requirements must be fulfilled in autonomous and resilient ways, such as failure detection, proactiveness, resource allocation, adaptability, and learning, inter alia [1,3,4].

While there is extensive literature combining MAS with sensor-based scenarios, we note that they are applied from an ad hoc perspective, where the proposed solutions are defined and designed specifically to the circumstances of the problem at hand [1,2,9,10]. Within the field of AI on MAS and agent reasoning for distributed problem solving, there is a mathematical framework that combines the fields of Artificial Intelligence and Operational Research called Constraint Reasoning (CR) that exploits the homogenisation and collaborative interaction of agents [2,19]. CR is a paradigm where the problem is modelled as a set of entities (variables) subject to constraints over and between them, where MAS is a suitable approach to model such problems in a distributed manner over real-world scenarios.

The open question we propose is how the field of sensor-based scenarios can draw on this approach to implement transportable solutions to different problems within the field, as well as to provide a basis for comparing different problems, experiments, and solutions on a common framework. Within this paradigm, Distributed Constraint Optimisation Problems (DCOP) arise to model and solve CP problems. DCOPs are the distributed generalisation of Constraint Satisfaction Problems (CSP) and provide a better solution flexibility, where these problems are solved by looking for the best possible solution rather than just satisfying the objectives or not. Problems such as graph colouring [20], multiple knapsack problems [21], task/resource allocation [22], and scheduling problems [23] have a distributed version, where agents are represented as vertexes, knapsacks, rescue vehicles, and planner devices, respectively.

However, the vast number of real-world scenarios with sensors tend not to be static, nor snapshot problems, so inherent dynamics as emerging modifications and/or disturbances that modify the problem are expected. In the same fashion, dynamics in distributed problem solving (including DCOPs) must be modelled to represent such events and behaviours. From a CR perspective, dynamics are linked to questions such as the increasing the problem (scalability) [21], reacting to failures (stability) [24], handling new knowledge or uncertainty [22], or considering future problem states [25], among others. To address this, Dynamic DCOPs emerge to represent and model these dynamics within the DCOP representation. In short, Dynamic DCOPs model problems that evolve over time, where conditions can change during the solving and, as mentioned below, agents and/or constraints are modified/added/removed to the problem. Besides, although the teamwork approach between agents to solve a problem can be cooperative or competitive, the literature considers cooperative behaviour as the default, regardless of the dynamics mentioned.d [26].

From our perspective, the DCOP system approach offers a systematic and applicable solution to many environments and problems, and this framework seems appropriate as a first approach to our open question, more specifically, the set of dynamic DCOPs. Despite this, dynamics over DCOPs are not always defined over the aforementioned criteria. Further, the literature also evolves and possesses certain heterogeneously. Terms, concepts, and dynamics modelling have no conventions to follow because they have been developed over time. Finally, several DSCP and DCOP surveys and studies have been conducted [26,27,28,29] but without a special focus on DCOPs with dynamics.

It is interesting to carry out a systematic study to understand the use and evolution of these systems that integrate distributed sensing and reasoning. Therefore, the aim of this paper is to identify, compile, and analyse the DCOP framework, specifically those that model dynamics over time, as well as to outline a state of the art that allows us to observe how involved proposals have been applied to real-world problems with the objective of discussing the feasibility and application of this framework to sensor-based scenarios. To do so, we conduct a Systematic Mapping Study (SMS) over the Dynamic DCOP research field. SMSs emerge as a powerful tool to sketch and structure in a broad manner the state of the art of a specific field. SMS have been used extensively in different fields related to computer science [30,31,32,33]. With this work, we aim to facilitate further research of such merge-distributed sensors and constrained-based research communities, providing a mapping of the current work and identifying trends, strengths, gaps, and opportunities.

The paper is described as follows: Section 2 introduces the Dynamic DCOP concepts and related background. Section 3 describes the method used for conducting the SMS. Section 4 responds to the proposed research questions. Section 5 discusses the applicability of our results to a sensor-based scenarios, and Section 6 concludes the paper.

## 2. Background

### 2.1. Multi-Agent Systems as Constraint Problems

Figure 1 summarises the agent functioning over a distributed sensor-based problem as a MAS. An agent is described as an entity that perceives its environment through sensors and acts upon that environment through actuators. The behaviour of an agent is composed of a continuous sensor–cognition–action cycle, which integrates the sensing process with the reasoning and execution of the action. Agents are capable of acting at any given instant according to their entire percept sequence observed to date, but not on anything it has not perceived yet. From a mathematical point of view, the agents act according to modelled beliefs, goals, and/or rules that maximise its utility, and its behaviour is described by the *function set* that map any given percept sequence to an action. When a problem is faced by a set of agents instead of one, it is defined as multi-agent. The multi-agent paradigm has additional design implications, such as orchestration, organisation, and communication. The nature of the environment is said to be *dynamic* if it can change while the agent is deliberating, so its world perception is continuously updating, as well as its plan and the inner state [4].

Constraint Satisfaction Problems (CSP) and Constraint Optimisation Problems (COP) are well-known frameworks for researchers seeking to solve problems by merging AI and MAS-based systems with their resolution through Operations Research techniques, such as constraint programming, mathematical optimisation, and decision analysis, among others. Despite their potential for modelling complex systems, CSPs and COPs are NP-hard problems and require a large amount of computing resources due to their high complexity [19]. Furthermore, some problems are not inherently centralised or are unsuitable to be solved by a single entity. Consequently, distributed versions of these frameworks emerged, either to share the computational load or represent problems composed of a set of smaller systems, in the same fashion as distributed sensor-based problems.

### 2.2. DCSP and DCOP

DCSPs and DCOPs are distributed extensions of CSPs and COPs [19]. Both are defined with the following tuple 〈A,X,D,F,α〉, where:$A=\{a_{1},\ldots ,a_{n}\}$ is a finite set of autonomous agents.$X=\{x_{1},\ldots ,x_{m}\}$ is a finite set of variables.$D=\{D_{1},\ldots ,D_{m}\}$ is a finite set of domains for the variables in **X**, where $\forall x_{i}\in D_{i}$.$F=\{f_{1},\ldots ,f_{k}\}$ is a finite set of functions, commonly called *constraints*. The set of variables relevant in an $f_{i}$ is the *arity* of the constraint.α:X→A is a mapping function where each variable x∈X is assigned to an agent α(x). An assignment γ is *complete* if all variables are assigned with a value.

Agents perform the value assignment only to the variables they can control through communication with other agents. In a DSCP, the goal is to find a complete assignment γ from α that satisfies all the constraints of the problem. DCOP is a generalisation of the DSCP framework [20,26,34], where constraints to the binary concept of satisfying is are to a cost, utility, or reward function. From this generalisation (using cost functions), an unsatisfying constraint in a DSCP is modelled as an infinity cost or otherwise a finite cost. Thus, a DCOP *solution* is a complete assignment where all constraints are satisfied. Accordingly, the goal of a DCOP is to find a complete solution Γ that minimises (by cost functions) or maximises (by utility or reward functions) the objective function of the problem, described as an aggregation over X:(1)$$\Gamma =\underset{x\in \Theta }{arg min}\sum_{f_{i}\in F}f_{i}\left(\gamma_{x^{i}}\right)$$
where Θ is the set of all possible solutions, or *state space*. In addition, a function $f_{i}$ is a *hard constraint* if its value is unfeasible when it is violated. Otherwise, when a constraint can be violated but incurs a penalisation is a *soft constraint*. It is important to note that DCOPs are NP-hard problems [20]. DCOP algorithms aim to solve them reducing certain sub-problems in polynomial time at least such as memory per agent (search algorithms) [35,36] or reducing agent network (inference algorithms) [34,37].

For a better understanding, the follow example is shown in Figure 2. A drone team carries out a resource allocation problem where a set of spatially located and heterogeneous tasks (imaging, tracking, delivering, etc.) must be performed by the drones in a distributed fashion. The allocation problem must consider different characteristics of the involved entities such tasks needed for their completion, available drone resources, distances to cover, and possible conflicts during allocation proposals by each drone. In our example, we have different roles for drones and types of tasks depending on their capabilities and objectives, respectively. A drone team comprise camera drones for imagery (blue), delivery drones for deposit goods (red), and complete drones (black) for imagery and delivery. Accordingly, tasks are divided by imagery (blue), video (green), or delivery (red). Tasks and drones are located spatially, but drones have a defined radio action that secures their position and makes them possible to recover if they fail or are out of battery. A possible version of modelling this problem would be the following. Agents, represented by drones, are assigned to tasks, representing variables in a DCOP. The domain of each variable or task are the map of all the possible drones that can perform such task. The allocation of a task to a drone can be measured as its utility. The utility can be modelled as a function based on the agents’ status and the constraint set of the problem. These constraints are related to several facts such as the agent capability (a drone is only capable of performing these tasks where its role matches with the tasks), distance (drones only perform tasks inside of their action radio), availability (a drone cannot perform two tasks at the same time), resources (such as drone speed or maximum weight capacity for delivery drones), priority (critical tasks must be performed before non-critical ones), and conflicts (drones must negotiate in order to allocate these tasks located in the overlapped action area), among others. An example of a soft constraint could be when a delivery drone decides to perform an imagery task with a low priority rather than other one with high priority. It is possible that there is a behaviour that is not desired but guarantees the task’s completion. On the other hand, drones that cannot perform delivery tasks or that are out of the action radio are examples of hard constraints. The goal in this scenario is to find an allocation of all possible tasks to drones where the overall utility is maximised.

DCOP algorithms are highly driven by the scenario. DCOPs are modelled in the same way as other MAS problems using graphs. Briefly, a MAS graph comprises nodes representing agents and edges as communication channels between nodes. In a DCOP problem, the graph not only define the agent interaction, but also how the algorithm coordinates its execution and models other relationships such as constraints or value assignments. Commonly, a DCOP graph representation can be classified into the following three types: constraint graph, pseudo-tree, and factor graph [26,29]. Figure 3 from Fioretto et al. [26] shows an example of the different DCOP representation.

A constraint graph is the most common DCOP representation where edges represent the constraints related to the involved variables. The number of variables connected by the edge determines the *arity* of the constraint. For a better representation, constraints with more than 2-arity are decomposed into binary constraints. A pseudo-tree graph is a spanning tree where nodes are ordered in parents, children, and leaves. This tree results from a depth-first search (DFS) from the elected node as root. Parent nodes are also ordered by priority for building the DFS tree without losing constraint information. these nodes connected as non-higher priority are named pseudo-parents and pseudo-children. A factor graph is a bipartite graph composed of two disjoint sets of nodes named as variable nodes and factor nodes. Edges only connect variables with factors. Variable nodes represent the set of variables, whilst factor nodes represent the constraint/cost functions of neighbouring variables, representing the constraint arity.

### 2.3. Literature Overview

Yokoo and Hirayama [27] published in 2000 a first review for CSP, including applications, algorithms and two proposed extensions for DCSP formalisation. Meisels [28] published in 2008 a book that introduces in detail search algorithms for DCSP and DCOP, including a study of their performance and different discussions related with agent communication, problems, trends, and challenges. William Yeoh sketched in their dissertation a taxonomy for classical DCOP algorithms. This taxonomy shows a basic DCOP classification based on solution quality, decentralisation type, and strategy with examples [38]. This taxonomy has been extended in different surveys. Fioretto et al. [26] and Leite et al. [29] included the agent synchronicity and added other strategies such as sampling and stochastic algorithms, respectively, with their examples.

Fioretto, Pontelli, and Yeoh’s survey performed a complete overview of the DCOP research field. The study delves into the different DCOP classifications and applications, with a description of the most representative algorithms of each DCOP family. In terms of solution quality, algorithms are classified as *Complete*, when the optimal solution can be reached or *Incomplete*, when non-optimal solutions are found (in exchange for decreasing computing and communication overload between agents). Regarding agents constraint information, DCOPs algorithms are *Partially centralised* when agents delegate constraint information to one agent for centralising local decisions (for avoiding conflicts), or *Fully decentralised* when each agent only has its own constraint information (for example, when privacy is key problem). In terms of algorithm execution, DCOP algorithms are *Synchronous* if agent operations and actions are strictly subject to the output of their neighbours, or *Asynchronous* when agent reacts exclusively from its perception of the problem. There are several more classifications such as agent/environment behaviour (deterministic or stochastic), algorithm strategy (search, inference…) and complexity measures (time, agent memory, agent communication…), among others. DCOP research field is a vast family comprising different variants and extensions [26]:*Asymmetric DCOPs*: Constraints with different costs depending on which agent controls a certain variable.*Multi-Objective DCOPs*: There is more than one objective function to optimise.*Quantified DCOPs*: Agents are competitive or not fully cooperative.*Probabilistic (Stochastic) DCOPs*: Stochastic cost functions to model non-deterministic environments.*Dynamic DCOPs*: DCOP changes over time due to dynamic events.

These families are not completely exclusive among them. Leite, Enembreck, and Barthès’ review [29] provides an overview and a comparative of different DCOP algorithms, focusing on algorithms from classical, asymmetric, and quantified DCOP families. Concerning agents’ behaviour, while it is true that MAS can be composed of both competitive and cooperative agents, in the Dynamic DCOP model, agents are fully cooperative and deterministic by definition. This paper belongs to this last family, and possible proposals that combine in a certain way Dynamic DCOPs with other categories are shown.

While these surveys developed provide an overview of the field, they do not consist of any systematic study but on manual literature reviews based on well-known algorithms. Indeed, these studies are focused on the DCOP field without focusing on the literature of any concrete sub-group or family, or even omitting them in some cases. Concretely, none of these studies base their purpose on Dynamic DCOPs, despite the fact that this field has been developed extensively for approximations of real-world problems. Assuming this, this paper aims to fill that gap showing the current overview of the Dynamic DCOP family. To do so, a systematic mapping has been conducted to obtain a corpus based on a refined method of searching, inclusion/exclusion criteria, and keywording-based classification.

### 2.4. Dynamic DCOP

As we introduced above, numerous distributed constrained problems modelled as DCOP possess inherent dynamics. Having said that, proposed algorithms must adapt to these dynamics due to the environment transform over time. To address this, the Dynamic DCOP model emerges to represent these problems. According to Fioretto, Pontelli, and Yeoh [26], a Dynamic DCOP is defined as a set of classical DCOPs $\left\{\delta_{1},\ldots ,\delta_{T}\right\}$ where each $\delta_{t}=\langle A^{t},X^{t},D^{t},F^{t},\alpha^{t}\rangle$ is a DCOP representing the problem at each time window *t*.

Essentially, a Dynamic DCOP can be sketched as a set of *n* DCOPs to solve sequentially. Accordingly, Dynamic DCOPs are also NP-hard because they require solving each DCOP in its respective time window. Nevertheless, this definition of dynamics is somewhat vague and must be described in detail for being modelled. Dynamics must involve one or several DCOP components, either any set of agents, variables, domains, and constraints of the problem, resulting in an adapted solution for the new configuration.

In our drone team example, dynamics can be represented as new drones incorporated to the team or those that become unavailable due to failure, tasks appearing or disappearing from the scenario, managing sub-teams of drones to perform a single task by a single agent, new types of tasks or drones, or restrictions modified based on uncontrolled events (weather conditions, human interaction, etc.), remaining resources of the drone team (battery, remaining payload, storage for imagery, etc.), or new distances from drones to tasks due to their reallocation. Figure 4 shows a set of snapshots in the Dynamic DCOP version for visualising several of the aforementioned dynamics. The first snapshot represents the initial state of the problem where a drone performs a delivery task. Suddenly, new tasks appear, and the previous one is relocated. Thus, the drone team updates its knowledge and faces the new problem. In the last snapshot, a new task appears in an overlapping area, but due to hard constraints (drones cannot perform tasks out of their range nor tasks unfeasible with their role), it is performed by the furthest drone.

The Dynamic DCOP field has been extended in various ways. Most well-known algorithms emerge from classical DCOP predecessors [39,40,41,42] or novel ones [22,43,44]. In addition, there are proposals in the literature based on existing algorithms adapted to handle the scenario evolution in a certain way [45,46,47]. Generally, all these aforementioned tend to inherit the characteristics of their predecessors and provide new properties in relation to the dynamics that are modelled in the problem. Some of these inherited characteristics are the solution quality, agent communication, or complexity. Another remarkable feature is the Anytime property, described as the ability of a concrete DCOP algorithm to provide or guarantee a solution regardless of its execution time [26]. On the other hand, a set of new properties arise from the dynamics to be considered in the problem as novel features in the Dynamic DCOP literature. Some of these features are related to problem stability, scalability, uncertainly, or fault-tolerance. Accordingly, these facts have been enriched and evolved the literature based on the needs of the moment. In fact, new families of algorithms have appeared in the literature as a cause of this evolution [17,25]. The purpose of this paper is to map the Dynamic DCOP literature according to the objectives and properties of the most representative contributions of this field to discuss how suitable they are as a framework to model dynamic sensor-based scenarios.

## 3. Method

The purpose of this paper is to visualise the Dynamic DCOPs landscape and identify trends and gaps, aiming to discuss the feasibility and applicability of this framework to sensor-based scenarios. We consider that the concerned literature is highly scattered, as is the terminology used for describing several components, features, and definitions, among others.

Systematic mapping studies are used to provide an overview of a software engineering field of interest [33]. SMS builds a classification scheme and structure of the field and provides a visual summary by mapping the results. SMSs are conducted following a process that includes quality techniques aimed to be repeatable and to minimise possible biases. SMSs have been used extensively in different computer science fields, including AI [48,49,50,51]. There are several guidelines in the SMS literature [33,52]. In fact, Petersen et al. [32] provides a set of updates conducting an SMS of existing guidelines. In the following section, we describe the used method, including research questions to answer in this paper and the obtained classification.

In order to discuss how distributed sensor-based problems can be addressed through the Dynamic DCOP model, we formulate the following main question: *How are Multi-Agent Systems through DCOPs applied in real-world and dynamic problems as sensors?* To do so, and with the best of our knowledge, we conducted a Systematic Mapping Study following the well-known guidelines [33] and updates [32]. SMSs are widely used to structure or provide an overview of a research area. In addition, mapping studies also aim to show research trends and gaps. From this question, we complete the following research questions (RQs):

### 3.1. Research Questions

This mapping study aims to identify, scope, and classify Dynamic DCOPs literature. Considering this, the following research questions (RQs) have been considered:**RQ1**: Which real-world scenarios and sensor-based problems have been considered to be modelled as Dynamic Distributed Constraint Optimisation Problems?**RQ2**: Which properties for Dynamic DCOP algorithms have been measured in terms of quality and performance (memory and communication)?**RQ3**: Are there any other properties that are relevant in these algorithms for sensor-based problems?**RQ4**: How has the Dynamic DCOP literature evolved over time?

These questions allow us to perform a mapping of the field without losing sight of which sensor-based scenarios they are applied to over the MAS research field in combination with the DCOP approach.

### 3.2. Search

According to Kichenham et al. [52] and Petersen et al. [32] we considered using the PICO approach (Population, Intervention, Comparison, and Outcomes) for performing the search-and-identify keywords.

*Population*: In our context, the population comprises DCOP scenarios in the multi-agent systems field.*Intervention*: In multi-agent optimisation, the intervention is highly driven by the scenario. So, we consider publications where dynamic scenario is defined or an algorithm that handle dynamics is proposed.*Comparison*: In this study we compared the fields and scenarios where Dynamic DCOPs are applied, the features according to the DCOP literature (quality, performance, memory, communication, etc.), and these features that are unmapped a priori but arise from the needs of the problem.*Outcomes*: For this study, we grouped the different candidates according to their features and predecessors. In addition, we drew a genealogical map of the different groups or families according to their evolution over time. Finally, we discussed the trends and gaps in the current dynamic scenarios.

Because we want to capture the field of study over time, we have the limitation of not being able to design a highly refined search. Concepts and terms evolve, and the proposed PICO enables us to define our search in a permissive way and refine our final candidates following the steps of the method. According to that, the keywords from the PICO are Distributed Constraint Optimisation Problems and Algorithms.

The search string, the selected databases, and the papers resulting from the search are shown in Table 1. The search strategy considers publications in the main repositories where the distributed problem solving with constraint-reasoning field is located. In addition, we add other two meta-repositories for possible candidates out of the focused repositories. We use RefWorks as our reference management tool to manage the large number of results and apply the following study selection.

### 3.3. Study Selection

Due to the relaxed criteria for the search phase, we applied a greater effort in the candidates’ selection. Accordingly, we applied the study selection in three phases. In a first iteration we excluded articles based on titles and abstracts. For the second iteration, we conducted a full-text reading. Finally, we applied quality assessment to both included and excluded papers and snowball sampling to the remaining included documents. Table 2 summarises the inclusion and exclusion criteria.

According to Fioretto et al. [26], we decided to include papers including and after the early algorithms scoped in DCOPs [20,34] and Dynamic DCOPs [22,40]. This decision prevented us from obtaining a large number of papers without losing potential candidates. However, as a safety measure, snowball sampling has no time restriction and enables us to identify possible out-of-range predecessors and identify the origin of the field to map. Included papers must contain at least an application of an algorithm when a dynamic scenario is proposed. Regarding exclusion rules, we avoided long publications such as books or thesis, and short papers such as extended abstracts or posters.

The evaluation was performed by the second and the third author as reviewers. The first author included a validation set composed of the Dynamic DCOP papers in the survey of Fioretto et al. [26]. These six papers [40,41,42,43,44,53] ought to be selected as included in the final selection of all authors. In addition, once each author selected the final set of papers from the full-text reading phase, and the two reviewers applied a set of combination rules to validate their selection with the first author. Decision rules were used by Petersen et al. [32] to validate paper selection and avoid possible biases. Table 3 shows our combination for decision rules. Finally, we performed snowball sampling to the final selection of papers. The process of searching and selecting papers is sketched in Figure 5.

### 3.4. Keywording and Classification

The data extraction was performed following the keywording technique proposed by Petersen et al. [33]. All authors read separately the abstract, introduction, and conclusions looking for keywords, concepts, and terminology that reflects the contribution of the papers. Then, the authors decided an inter-rate agreement to validate keywording and avoid possible biases.

For the data extraction and classification, a form was defined, directed by the resulting keywording and the RQs. The data extraction template is shown in Table 4, where rows are grouped in different classification categories. These categories are the following:(1)*General*: Involves the basic paper information to classify and its scope. It includes the venue-related info, the field of application where the study is scoped, and the problem definition according to a MAS scenario. The latter should include information about which entities are represented as agents and other aspects such as communication, behaviour (competitive, selfish, collaborative...), and knowledge representation (total or partial), among others.(2)*Modelling*: Covers the related DCOP characteristics according to [26], which include the graph representation, complexities, and related properties. In addition, we identified the dynamics that were addressed by the studies to justify the modelling as a Dynamic DCOP.(3)*Related Work*: For each included study, we have extracted information on predecessor studies, related work, and selected candidates (if any) to cross-check results during the evaluation/experimentation sections.

### 3.5. Quality Assessment and Validity

The full-text phase showed us further publications out of scope. In this phase, the quality assessment was performed to both included and excluded papers by the three authors. On this set, the following questions were suggested to evaluate the quality.

Does the proposed scenario/solution identify or model dynamics in the scenario or environment?Is the background/related work of the paper clearly scoped to DCOP research field?Has an evaluation/experimentation been conducted?

To evaluate and validate our search, we defined a set of papers extracted from Fioretto et al. survey [26] that authors should accept ought to be included during the process [25,40,41,42,43,44,53,54]. The full text excluded papers include works with a certain degree of duplication. This is the case for some works whose contribution was an extension of previous studies in terms of experimentation or contextual change, rather than a new proposal that evolves from previous work.

In order to minimise possible biases during data classification, one author extracted the data from accepted papers whilst the other two reviewed the extraction. Due to the time frame and the heterogeneity of the field under study, we assume the risk that the accepted papers could in some cases not provide all the required data during their extraction. Otherwise, this would imply such a high restriction that we would discard a large number of papers that should be included. Therefore, we discussed those papers that might have some data gaps or were not very descriptive in some respects, as long as they passed the quality questions described above.

## 4. Results

### 4.1. Scenarios, Topics and Venues (RQ1)

#### 4.1.1. Scenarios

Figure 6 shows the number of publications of interest for this study from 2005 to the 2020. We notice that in certain clusters of accepted DCOP papers in concrete topics, only one per field is aligned to our research questions and proposes an algorithm based on Dynamic DCOP. Below, we list these scenarios considered in the included papers, as well as the description of how the dynamics are modelled:
**Mobile Sensing Agents** [17,18,45,55,56,57,58] or MSTs consist on teams of unnamed vehicles with different physical sensors that collect their environments to track different targets, map the environment, covering an area, identify events, or perform collective actions on the ground, among others. In this scenario, sensing agents must solve two sub-problems: exploration and exploitation. Exploration tasks focus on analysing the environment and searching for goals and tasks, while exploitation is about taking actions (in relation to their goals) that maximise their utility. In addition, algorithms must consider the balancing of both tasks. The dynamics that change the DCOPs to be solved are reflected in the modifications in agent communication during exploration (agents appearing and disappearing from the range of their moving neighbours) and tasks appearing and ending during exploitation. Zivan et al. proposed the DCOP_MST model, which works with different incomplete local search-based (MGM, DSA) and inference DCOP algorithms (Max-Sum) for simultaneous exploration that updates the domains and constraints of neighbouring agents and exploitation to solve the new setup accordingly [17]. Taylor et al. proposed a framework where incomplete algorithms (DSA and Max-Sum) allow agents to interleave between “stay” and “explore” states, depending on their attempt to maximise their cumulative reward on their available actions or deciding on exploring to sample and update their reward function given a time horizon [18,55]. Hosseini and Basir proposed a different approach through partitioning of the main DCOP as a target for a sensor allocation problem on different sub-problems such as regional DCOPs in a hierarchical structure, where sensors and targets are mobile, so they traverse between regions modifying the involved DCOPs [45]. Mailer et al. proposes a novel solving proposal through the transformation of Dynamic DCOP onto a thermodynamic system and applying its three laws. This model is evaluated over a simulation of a sensor network problem where dynamics are represented as targets being displaced over a grid being detected by near sensors [58].**Disaster Response** [21,22,41,46] is focused on environments where autonomous agents form teams to face emerging rescue tasks associated with natural disasters and human emergencies, among others. In these scenarios, time is highly constrained, so DCOPs are expected to be solved as fast as possible to allow agents to be available for further tasks. Indeed, there are research projects such as Robocup Rescue (https://rescuesim.robocup.org/, accessed on 14 May 2021) that provides a framework for evaluation and benchmarking of MAS-based response plans to act in real natural disaster situations. Scerri et al. and Ramchurn et al. developed LA-DCOP (Low-communication Approximate DCOP) [22] and Fast Max-Sum (FMS) [41] as inspired faster versions of DSA and Max-Sum algorithms, respectively, for task allocation. Both algorithms greatly speed up the resolution of DCOPs in reduced time windows through techniques that reduce the amount of message passing between agents during resolution, to the detriment of finding optimal solutions.**Distributed Web Services** [59,60] refers to IoT-based elements represented as distributed and cooperative agents in control of their respective web objects and services. The problem to solve is how to perform a continuous distributed orchestration and/or a load balancing of the involved services, where dynamics are represented as variations over time in the demand for such services.**Smart Grids** [23,61,62] scenarios, where agents manage and control different elements from the electricity grid such as generators, dispatchable loads, feeders, etc., and dynamics are represented as variations in the energy supply and demand in consecutive time windows. Fioretto et al. define the Distributed-EDDR problem to model the Economic Dispatch and the Demand Response as two parallel sub-problems being merged in order to solve it through a DPOP-like algorithm solver [61]. Lezama et al.’s approach was based on the resolution of sequential canonical DCOPs with policies that enable agents to maintain or update their domains and constraints between consecutive time windows [23,62].**Traffic flow** [63,64,65] consists of performing plans and scheduling systems over transiting vehicles in order to avoid jams and/or bottlenecks. In these, studies agents are in control of a set of different elements such as traffic lights on crossroads [63,65] or platforms from railway systems [64], where dynamics are represented as the variations in the traffic density from circulating vehicles.**Other scenarios** were Dynamic DCOPs are modelled include collective autonomous vehicles for collision avoidance [66], human resources reorganisation such as scheduling elective surgery for emergencies in hospitals [67], distributed controlling of IoT devices and sensors for minimisation of energy consumption in smart homes [47], and dynamic social simulation through law enforcement problems and market-based mechanisms for dynamic task allocation to agents [68].**Proposals without scenarios** [24,25,39,40,42,43,44,53,54,69,70,71,72,73,74,75] are the majority of the studies found. These studies refer to possible applications but are merely introductory examples without the context of a specific scenario. This evidences to some extent a gap between the extensive study of the field and its application to real-world problems.

#### 4.1.2. Venues of Publication

One of the concerns is to identify and analyse the sources of these articles. According to Fioretto et al. [26] and to the extent of our knowledge, the first algorithms relevant to our study were published in 2005, and since then, publications have become more popular over time.

Regarding the venue type, Figure 7 provides the number of articles that were published in different venues, and Table 5 thprovides e top used venues in the field. In total, 82% were published in proceedings from conferences and linked workshops, while 18% were published in journals, with the AAMAS conference and the Lecture Notes on Artificial Intelligence/Computer Science being the most relevant.

### 4.2. Algorithms Properties (RQ2)

Table 6 classifies the accepted papers according to the common properties from DCOP algorithms [26]. The table includes extracted information about how agent communication is established (as described in Figure 3), its optimality (whether solutions are complete or incomplete), and complexity analysis provided by the studies. We also identified those proposals that designed an extension of the Dynamic DCOP model and novel frameworks that introduce an off-the-shelf classic DCOP to a dynamic problem. One of these model extensions is multi-objective optimisation, where the Pareto value on the Optimal column is used. Therefore, a solution is said to be Pareto optimal if and only if there is no other solution that improves at least one objective function without decreasing the cost of the remaining ones. Please refer to Section 4.3 and Section 4.4 for further explanation of frameworks and model extensions.

Concerning the complexity columns, we only extracted data of studies that provide a clear analysis of such information. The dotted marks indicate studies where complexity issues were assumed from the algorithm predecessor, or where there are multiple algorithm versions, such that diverse complexities were inherited. Is the case of MST and DER algorithms with versions from DPOP, AFB, and Max-Sum algorithms [17,18,23].

### 4.3. Features (RQ3)

The following list describes the different improvements and advanced properties identified from selected Dynamic DCOP algorithms according to the scenario needs and restrictions:**Scalability** [21,22,24,41,60,66,68]: by definition, DCOPs and Dynamic DCOPs are both NP-hard, so algorithm scalability persists as a critical challenge in MAS research. LA-DCOP, FMS, and BnB-FMS (Branch and Bound FMS) outperform other algorithms such as DSA or S-DPOP in both global utility and time execution with a high growth of tasks and agents due to their mechanisms for reducing communication between agents (number of messages and size, respectively) [22,24,41]. Ferreira et al. designed the Swarm-GAP as a heuristic DCOP algorithm based on ant colony optimisation able to achieve similar performance to LA-DCOP in experiments with abstract simulators (up to 5000 agents) and the RoboCup Rescue Simulator, but lowering the communication and computation effort [21].**Stability** [24,40,42,44,47,53,60,74]: Dijkstra defined this property in dynamic systems as the agent’s ability to react and adapt as fast as possible to emerging changes in their environment [76]. DCOP literature adopted this definition to represent the property of an algorithm to minimise the computation required to find a solution each time the problem changes [26]. Petcu and Faltings modelled the Self-Stabilization feature for DPOP agents (S-DPOP) based on the properties of convergence (a stable solution is guaranteed in a finite number of steps) and closure (agents do not change the solution if the scenario does not change either) [40]. Further Dynamic DCOP research based on stability has emerged from this definition, and S-DPOP has become a well-known algorithm for comparing experiments. Yeoh et al. adopted this feature for ADOPT algorithms (any-space ADOPT and any-space BnB-ADOPT) [42] and Macarthur et al. for FMS algorithms (Bounded FMS) [74]. Petcu and Faltings designed RS-DPOP, an improvement of the S-DPOP algorithm based on deadlines in agents on the commitment of their variables. Billiau et al. proposed the SDBO algorithm considering stability from another approach based on fault tolerance, ensuring problem resolution if there are changes in the agent network or if agents fail [44].**Privacy** [59,67,72]: Some distributed scenarios require the preservation of data privacy between those entities managed by agents (Smart buildings, IoT, Web Services, Networks, Competitive scenarios, etc.). In these scenarios, agents do not share critical information with neighbours, but mainly the costs or utility associated with their actions or its local variables assignment. In these scenarios, the concept emerges of intra-agent variables as private and inter-agent variables allowed to be shared, with their respective constraint relationships. Wahbi et al. modelled Data Centres as agents with information related to Virtual Machines, energy consumption, and energy cost as intra-agent variables [59]. Khanna et al. proposed a degree of dissatisfaction value shared between agents based on private intra-agent constraints costs [67]. Billiau et al. modelled privacy as Asymmetric DCOPs because they are suitable when constrained agents incur different costs for a joint action in order to preserve their local utility [26,72].**Anytime** [21,24,41,44,56,66,70,73]: A DCOP algorithm is said to be Anytime if it can return a valid solution (but incomplete) even if the agents are interrupted at any time before the algorithm terminates. Anytime algorithms tend to converge for solutions with better quality as they keep running. Anytime approaches on incomplete algorithms are interesting on large DCOPs, with high frequency of emerging dynamics and uncertain environments. While Stability and Anytime are characteristics related to fault-tolerance, they are not interchangeable. Anytime mechanisms aim to guarantee a solution, either complete or not, whilst stability seeks to minimise efforts between one solution and the next one when the problem changes. In other words, the Anytime property of a Dynamic DCOP is applied over each single DCOP instance, and the stability property states is applied between instances.**Learning mechanisms** [43,55,63,65]: Machine learning techniques are used on agents to find policies that maximise their utility during solving. Nguyen et al. modelled the Markovian Dynamic DCOP (MD-DCOP) as a sequence of classic DCOPs where transitions are represented as a Markov Decision Process (MDP) and proposed a distributed version of the Q-Learning algorithm [43]. Machine learning agents on Dynamic DCOPs based on MDPs and Q-learning algorithms were also approached by [55,63,65].**Multi-Objective** [69,71,72,75]: Okimoto et al. modelled the *Dynamic Multi-Objective DCOP (DMO-DCOP)* as a sequence of static MO-DCOPs [69,71]. As explained before, Multi-Objective Optimisation consists of solving DCOP problems with more than one objective function that must be optimised simultaneously, with the possible existence of trade-offs among objectives. According to this, there is a possibility that maximising all objectives in the same solution is unfeasible. Instead, MO-DCOPs are solved when a Pareto Front is found, as the set of Pareto Optimal solutions that are dominant for their respective objective functions.**Uncertain management** [55,57,63]: Certain scenarios model dynamics in such a way that agents must act under uncertainty. Taylor et al. coined the term “team uncertainty penalty”, where joint decisions by a team of agents acting on dense graphs with high constraints variability lead to a significant degradation in team performance [55,63]. Fransman et al. represented uncertainty as the positional error between the current and the estimated position of a target being tracked by an agent [57].**Proactive** [25,54] By definition, intelligent agents with goal-driven behaviours are able not only to react to changes but also to take initiatives beyond such reactions in response to their environment. Hoang et al. introduced this concept and identified a new family of DCOP problems as Proactive Dynamic DCOP (PD-DCOP). Proactiveness is connected to the question of how agents can predict future changes assuming some degree of uncertainty. To do so, they extend the DCOP model with a finite horizon as time steps where agents can change the value of their variables, a switching cost of such values between time steps, and a set of transition functions that randomly changes the values in successive time steps. The goal of a PD-DCOP is to find a sequence of assignments for all variables over such horizon. Given this extension, they designed both complete and incomplete algorithms based on DPOP and S-DPOP predecessors [25,54].**Framework** [17,43,55,62]: We identified a set of papers that propose diverse structures or problem definitions to handle Dynamic DCOPs, where a off-the-shelf classic DCOP is assembled instead a pure novel algorithm. The DCOP_MST model introduces changes in constraints through the inclusion of dynamic domains [17]. DCEE problems were defined as a framework to introduce and coordinate two algorithms for exploration and exploitation ofsensing agents [55]. Lezama et al. proposed a framework where classic DCOPs on consecutive time windows are connected by an inter-phase operation where a set of helper agents updates information from dynamics in parallel with the current state of DCOP agents [62].

### 4.4. Dynamic DCOP Family Tree (RQ4)

Figure 8 and Figure 9 summarise the roadmap of the Dynamic DCOP field. DCOPs are problems that distributed agents solve in many different ways. Mainly, DCOP algorithms are grouped by search-based, inference-based, and heuristic-based algorithms. Search-based algorithms are based on the use of search techniques to explore the space of possible solutions, mainly based on best-first and depth-first search. Inference-based algorithms are derived from dynamic programming to solve mathematical optimisation problems by breaking them down into simpler sub-problems. Heuristic-based are incomplete approaches focused on fast approximations to solutions.

Throughout the development of the field, different problems have arisen to be transported to the Dynamic DCOP model, aimed at modelling and incorporating the peculiarities of specific scenarios. This is the case of MST and DCEE problems for mobile sensing agents [17,55], or DER and DEEDS problems for smart grids [23,61]. On the other hand, there are several works that have directly extended the model. These extensions are either inherited from other DCOP families or modelled as new functions and parameters to represent certain features described in the previous question:**Multi-Objective Dynamic DCOP** with algorithms such as DPR, DMOBB, and SSBDO [69,71,72]. Clement et al. formalised first the DMO-DCOP problem as a sequence of a set of MO-DCOPs in the same fashion [71]. In their work, dynamics as changes in successive DCOPs were connected to the changes in the number of objective functions, whilst the number of agents, domains, and costs did not change. Billiau et al. extended the SBDO algorithm, where dynamics are represented as changes in the constraint set of the scenario, to further belongs to the Asymmetric DCOPs model [72]. Agents in asymmetric problems are defined as selfish or untrustworthy but are still cooperative. Thus, agents may associate different costs depending on which agent they share their information with.**Probabilistic (Stochastic) Dynamic DCOP** that merges dynamics with stochastic (constraints) and machine learning mechanisms (distributed Q-Learning) from Probabilistic DCOPs where the MD-DCOP is placed. [43]. Dynamics here represent changes in the cost function from agents, whilst agents, constraints, and domains remain unchanged across all the successive problems.

**Proactive Dynamic DCOP** extends the Dynamic DCOP with the definition of horizon as a set of time steps where agents can change the value of their variables, with the aim to find a sequence of current and future assignments for such variables [54]. Further, Hoang et al. defined the *Infinite-Horizon PD-DCOP* (IPD-DCOP) model, which extends PD-DCOPs to handle infinite horizons through MDP [25]. The main idea is that the Markov chains on proactive problems will converge to a stationary distribution at some time step during the DCOP set sequence (either before a finite or infinite horizon). In these proposals, the dynamics are reflected exclusively in the change of the agents’ cost functions.

As illustrated in the roadmap, we observe that most studies are defined as an extension or improvement of well-known classical DCOP algorithms—mainly DPOP, Max-Sum, DSA, MGM and ADOPT. Therefore, few studies have iterated on pre-existing dynamic DCOP algorithms. This evidences the lack of comparison between different dynamic algorithms, where experimentation is mainly based on corroborating the improvement over predecessors. Uniquely, S-DPOP is the most widely used and referenced algorithm in the literature, comprising the origin of several families and algorithms listed above.

## 5. Discussion

In this section, we discuss the most relevant results obtained and respond to how this field is applicable to sensor problems, along with possible biases and flaws during this study.

### 5.1. Applicability of Dynamic DCOPs to Sensor-Based Problems: Research Gaps and Opportunities

#### 5.1.1. RQ1

As we identified in the study, IoT-related scenarios are widely assessed through Dynamic DCOPs, especially with Mobile Sensing and Disaster Response scenarios. The algorithms for Mobile Sensing Agents exploit exploration and exploitation sub-problems iteratively. Sensors update their information in the exploration phase and then proceed to solve the problem in the exploitation phase. This update includes all the changes of emerging dynamics such as their internal state, inclusion/removal of neighbouring agents, failure notifications, network changes, detection of new targets or tasks, and so on. Disaster response can be considered as a special group related to the first one, in which a very rapid resolution of the problem under extreme conditions and changes in the environment is a priority. The entity of unnamed vehicles (UAV for Aerial ones, such as Drones) controlled by the agents emerge in these scenarios [9,17,22]. Algorithms in this group are able to be very reactive and provide solutions in a very short time due to the low effort they exert in execution time as well as in their communication.

Although all the studies we have found refer to their usefulness in different problems to a greater or lesser extent, we have found a large number of these studies that do not apply to any specific scenario. As far as we understand, these models have a strong mathematical basis, and their application to the real world has been built up over time in the literature. However, we believe that this may be salvageable as the community itself has developed benchmark simulations such as RoboCup that allow for experimentation and comparison within a common framework. We consider that the use of these tools should be extended, because they allow a better modelling and application of dynamic sensor-based scenarios as constraint-based problems.

#### 5.1.2. RQ2

We have observed shortcomings in the analysis of complexity in many proposals. This is partly justified since novel proposals are designed on the basis of their predecessors, which inherit their complexity. Others perform such analysis based on the results during experimentation. In any case, many studies do not discuss aspects related to the problem growth, nor the communication load of the agents.

The trend of dynamic algorithms is focused on reducing the growth of the message size or the number of messages, where the incomplete approach is the most extended. Indeed, these algorithms are capable of obtaining optimal solutions if certain conditions are met. For example, Max-Sum and variants provides the optimal solution if their factor graph (represented as a disjoint set of tasks and agents) is acyclic [41,74]. In addition, algorithms that are focused on scalability such as LA-DCOP, Swarm_GAP, FMS, and BFMS are the fastest approaches [21,22,41,74]. In complete approaches, although they are in the minority and require more computational effort, these difficulties can be overcome under concrete conditions. For example, the induced width *w* is a parameter that determines the exponential complexity of DCOP agents organised as pseudo-tree graphs: DPOP-based algorithms inherit a complexity of $O(d^{w})$ in time and message size, where *d* is the size of the largest domain. In return, the number of messages grows as O(n), where *n* is the number of variables in the problem [40,53].

#### 5.1.3. RQ3

We have observed that algorithms based on dynamic DCOP pay special attention to those features that offer good performance in terms of solving, time, and size. Concretely, algorithms that consider Anytime and Stability mechanisms guarantee the problem solving in different contexts and time frames, either by providing refined solutions during solution improvement, as well as ensuring that the solution obtained is valid and remains valid until the environment changes, respectively. This approach is an excellent fit in sensor-based scenarios where low response latency and high adaptation to emerging dynamics are expected to be combined.

On the other hand, as mentioned in the discussion of RQ1, a great set the studies lacks a proper application to real-world problems and simulations. To the extent of our understanding, a greater effort on the use of benchmark simulations and frameworks should be made, especially in these studies where its objective is mainly focused on scalability, uncertainty, and scenarios with a high frequency of dynamics. Further, simulations on sensing agents in dynamic scenarios are quite useful as a first approximation for further IoT-based scenarios (mainly proofs-of-concepts, prototypes, and preliminary results). Simulations and distributed IoT share many problems, but the first are more convenient to work with: they require quite fewer physical devices, no specific materials are required, they are less less expense (including breakdowns), they are easily transportable, and they can be parallelized or scalable, inter alia.

#### 5.1.4. RQ4

Most of the works are based on the outperforming of classic DCOP predecessors rather than comparisons with dynamic ones. Although S-DPOP has been the most referenced and used dynamic algorithm in experiments, there is a lack of experiments that compare the performance and other aspects between the proposals. Considering that DCOPs are NP-hard, there is a trend towards the development of incomplete and Anytime algorithms. However, few consider the scalability aspect nowadays, as there is little evidence in studies when the number of agents is increased, especially in complete algorithms.

The most recent proposals (from 2015 to 2020) are the most appropriate to adopt today as this set has been applied in various IoT- and sensor-based scenarios [17,46,56,57,58,59,62] and include the Stability and Anytime mechanisms spread over different algorithms [42,47,56,62,68] that provide robustness on environments with a high frequency of dynamics.

### 5.2. Possible Biases or Flaws

As mentioned in the introduction to this paper, dynamics over DCOPs are not always defined over a common criterion, due to the sparse evolution of the literature where terminology is heterogeneous and has developed over time. Dynamics in Dynamic DCOP refers to scenarios that change over time based on the addition/removal of goals, agents, and/or constraints, not on the change of agents’ behaviour, or other heterogeneous definitions from the MAS field. In other words, Dynamic DCOPs are represented as a succession of classical DCOPs that must be solved consecutively in a given time window. Dynamic DCOP is a term coined subsequently and we previously detect some pre-2007 algorithms which, although part of the model was not self-identified within this model. Indeed, the word “Dynamic” has a very broad meaning and is used in different but related terminology, such as *Dynamic Programming* on Classic DCOPs. For this reason, we decided to do an extensive search omitting this word in our search string.

Additionally, papers that do not include the "DCOP" term are out of the first search of this study. However, we know that there are publications that obviate these term, or use other variants such as Distributed “Combinatorial” or “Satisfaction” Constraint Problems [40,74]. To minimise this problem, we have done backward snowballing to try to find publications of interest that escaped our original query. We also carried out a Quality Assessment, where we established a set of well-known papers that our process must include [26]. The inclusion of the word “algorithm” has allowed us to eliminate early on studies with the same acronym as DCOP but with different definitions (Civil Engineering, Medicine, Energy, etc.), as well as papers that superficially mention the topic but do not provide any proposal, without affecting the number of accepted papers. Moreover, the inclusion of meta-search engines such as Scopus and Web of Knowledge, as well as snowballing sampling, has allowed us to obtain papers that might be outside those already found in ACM or IEEE and to reinforce the inclusion of potential studies that might have been missed during the search.

The final queries were obtained when the successive iterations comprised a sufficiently high acceptance rate that implied both a minimal modification in the papers obtained and the same papers accepted after applying the inclusion/exclusion criteria. A large number of papers have been discarded due to an unclear definition and modelling of the problem dynamics. Another important reason for discarding has been studies with questionable experimentation and evidence. To mitigate this case, an evaluation criteria described in Section 3 was established.

## 6. Conclusions

By understanding agents as a continuous cycle of sensor–cognition–action, the usage of Dynamic DCOPs is an actual, powerful tool for changing environments with distributed and decentralised sensors. Through this framework, heterogeneous sensors can be controlled by agents that allow the standardisation of information and thus orchestrate their actions over time. Given the scattered study of this field, and to understand how we can apply the different proposals that compose the literature in our context, a systematic mapping has been conducted. Our results show that many proposals have been developed in scenarios related to IoT and distributed sensors, including Mobile/Wireless Sensors, Disaster Response, and Smart grids/homes, among others. Based on these scenarios, proposals have been designed to improve the performance of different aspects, mainly in the communication effort between sensors as agents. Because distributed optimisation problems are exponentially growing, the tendency is to opt for algorithms that provide sub-optimal solutions, or complete if and only if the size of the problem and the agents’ disposition allow it in a reasonable time. Other trending properties are those that guarantee the solution over time, either with mechanisms that provide a valid solution at any time, or the stability of the solution when faced with emergent dynamics. Other trending properties are those that guarantee the solution over time, either with mechanisms that provide a valid solution at Anytime, or the Stability of the solution when faced with emerging dynamics. These studies include other aspects such as scalability and the development of frameworks that allow the integration of classical off-the-shelf DCOP algorithms in a dynamic environment. Although we have observed certain shortcomings in several studies included in relation to their lack of application to real-world problems and certain aspects of their algorithmic analysis, there is evidence of a high applicability of the dynamic DCOP model to sensor-based scenarios.

A more in-depth review of algorithms, their methodologies and techniques used to solve problems that focus on mobile sensors and disaster response scenarios, and an experimental approach to the use of physical sensors and IoT in agent-controlled smart homes implementing Dynamic DCOP algorithms are proposed for future work.

## Figures and Tables

**Figure 1 sensors-21-03807-f001:**
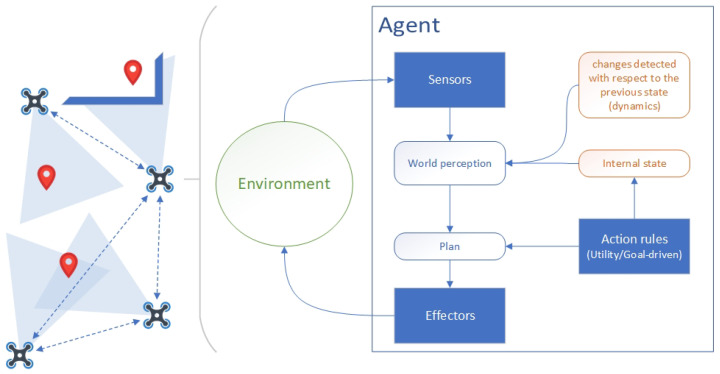
Hide-and-seek problem with a sensor-based system as a MAS and an example of conceptual agent reasoning based on maximising utility according to its goals [3,26].

**Figure 2 sensors-21-03807-f002:**
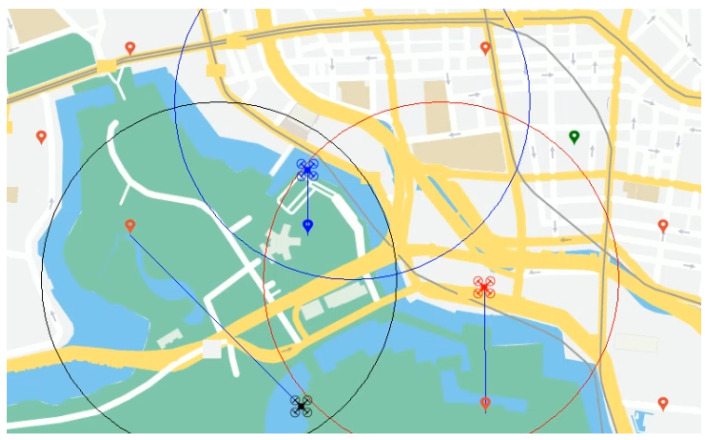
Drone team allocation problem example.

**Figure 3 sensors-21-03807-f003:**
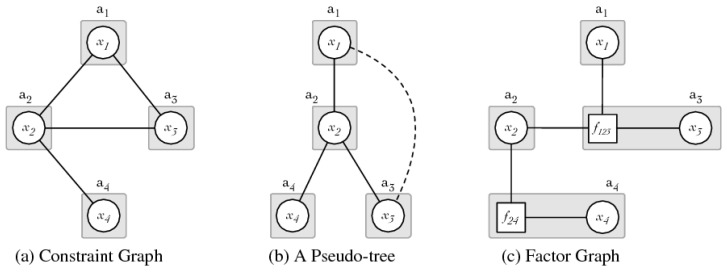
DCOP graph representation from Fioretto, Pontelli, and Yeoh’s survey [26].

**Figure 4 sensors-21-03807-f004:**
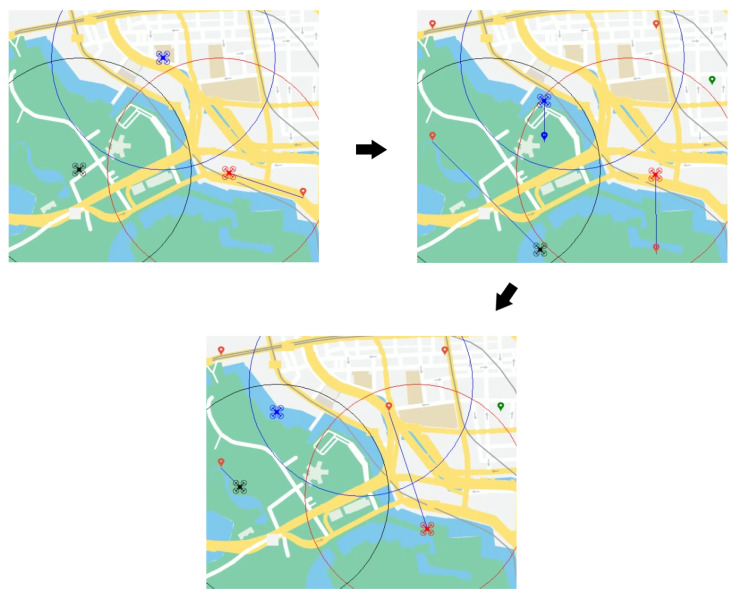
Drone team allocation problem as Dynamic DCOP.

**Figure 5 sensors-21-03807-f005:**
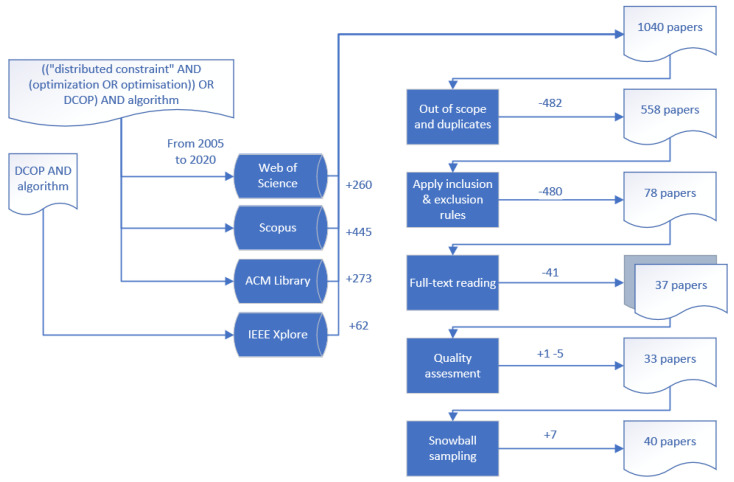
Search and study selection processes.

**Figure 6 sensors-21-03807-f006:**
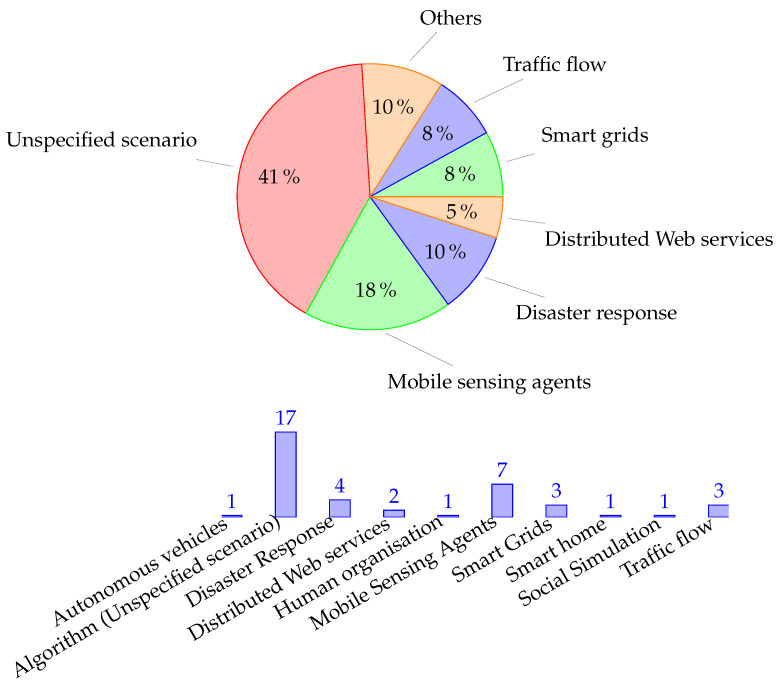
Distribution and summary of topics and scenarios distribution.

**Figure 7 sensors-21-03807-f007:**
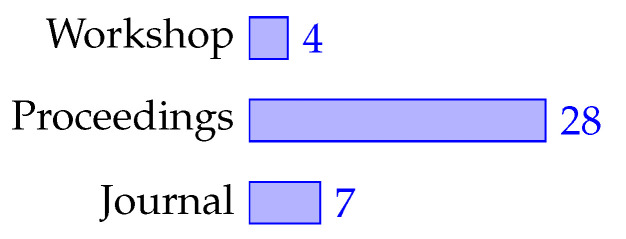
Venue types: Workshops [21,42,69,74], Proceedings [17,18,22,24,25,39,40,43,44,46,47,53,54,56,57,59,60,61,62,63,64,65,66,67,70,71,73,75], and Journals [23,41,45,55,58,68,72].

**Figure 8 sensors-21-03807-f008:**
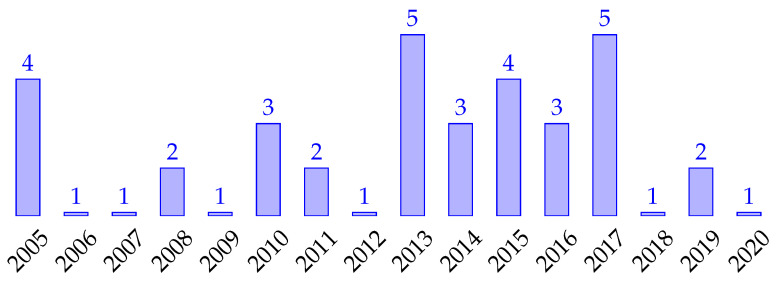
Distribution of primary studies by year.

**Figure 9 sensors-21-03807-f009:**
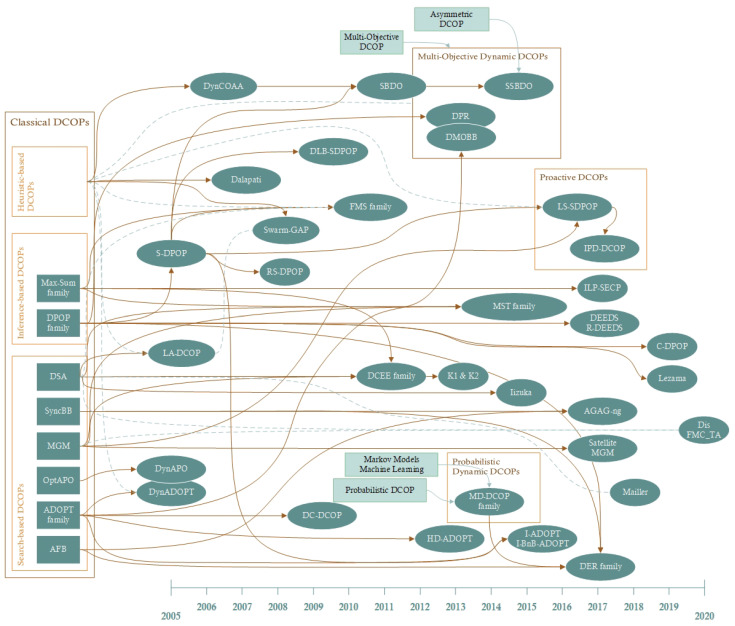
Roadmap of Dynamic DCOP literature from selected papers. Brown arrows represent connected proposals evolved from their predecessors. Dashed green lines represent only these algorithms used on experimentation for comparison.

**Table 1 sensors-21-03807-t001:** Databases, Search strings and publications per DB.

Database	Search String	Publications
IEEE	DCOP AND algorithm	69
ACM Inspec Scopus	((“distributed constraint” AND (optimisation OR optimisation)) OR DCOP) AND algorithm	285 176 378

**Table 2 sensors-21-03807-t002:** Summary of the inclusion and exclusion criteria.

Criteria	Rules
*Inclusion*	Papers in the field of MAS and DCOPs. Problem scoped in a dynamic scenario as consecutive DCOP problems. Papers published from 2005 to 2020, both inclusive. Papers with at least one of the following: - Algorithm proposal - Algorithms benchmark/comparison
*Exclusion*	Papers not in English. Papers not accessible in full text. Duplicates. Books and thesis. Short papers (less than or equal to 4 pages). Posters and dissemination work.

**Table 3 sensors-21-03807-t003:** Decision rules (included in bold).

	Author 1		
	*Included* (A)	*Doubt* (B)	*Excluded* (C)
**Authors 2 and 3**	**AAA**	**BAA**	**CAA**
	**ABA**	**BBA**	CBA
	**ACA**	BCA	CCA
	**AAB**	**BAB**	CAB
	**ABB**	BBB	CBB
	ACB	BCB	CCB
	**AAC**	BAC	CAC
	ABC	BBC	CBC
	ACC	BCC	CCC

**Table 4 sensors-21-03807-t004:** Data extraction template.

Data Item	Value	RQ
*General*		
Article	Bibliography ID	-
Title	Article name	-
Authors	Authors set	-
Year	Publication date	RQ4
Venue	Publication venue	RQ1
Area	Field application	RQ1
MAS problem	Agents and problem definition	RQ1
*Modelling*		
Graph	Constraint/Pseudo-tree/Factor graph	RQ2
Solution type	Complete/Incomplete	RQ2
Complexity	Time execution complexity	RQ2
Messages	Number of messages complexity	RQ2
Memory	Agent memory complexity	RQ2
Dynamics	Set of identified dynamics to model	RQ1
Properties	Set of highlighted properties	RQ3
	extracted from keywording	
*Related Work*		
Related	Set of related DCOP works	RQ4
Evaluation	Set of compared DCOP algorithms	RQ4

**Table 5 sensors-21-03807-t005:** Targeted venues.

Rank	Venue	Number	Studies
1	AAMAS	13	[17,18,22,25,39,43,54,58,61,63,65,73,74]
2	LNAI/LNCS	6	[21,44,47,59,69,71]
3	WI-IAT	4	[42,53,60,67]
4	AAAI	2	[24,40]

**Table 6 sensors-21-03807-t006:** Dynamic DCOP properties from SMS included papers. Optimality and Graph representations and Proof of time and spatial complexity such as Exponential, Polynomial, or Linear growing from algorithms are included.

Reference	Algorithm	Family/ Framework	Graph Topology	Optimal (Complete)	Proof of Complexity Analysis		
					Time	N° Msgs	Msg Size
[21]	Swarm-GAP		Constraint	✗		Linear	Linear
[22]	LA-DCOP		Constraint	✗	•	Linear	Linear
[23]	DER	✓		✗	•	•	•
[24]	BnB-FMS		Factor	✗	Exp.	Poly.	Linear
[25]	PD-DCOP(LS-SDPOP)	✓	Pseudo-tree	✓	Exp.		
[39]	DynDBA/DynAPO		Constraint	✗/✓		Poli/Linear	
[40]	S-DPOP		Pseudo-tree	✓	Exp.	Linear	Exp.
[41]	FMS		Factor	✗	Exp.	Poly.	Linear
[42]	I-BnB-ADOPT		Pseudo-tree	✓	Exp.		
[43]	MD-DCOP	✓		✗			
[44]	SDBO		Constraint	✗	Exp.	Exp.	Linear
[45]	HD-ADOPT/HD-DBA		Pseudo-tree	✗	Exp.	Poly.	
[46]	Iizuka et al.		Constraint	✗			
[47]	ILP-SECP		Factor	✗	Linear	Poly.	
[17]	MST	✓		✗	•	•	
[53]	RS-DPOP		Pseudo-tree	✓	Exp.	Linear	Exp.
[54]	IPD-DCOP	✓	Constraint	✗	•		
[55]	DCEE	✓		✗			
[18]	MST_Max-sum		Factor	✗	Linear		
[56]	Sat-MGM		Constraint	✗	Poly.	Poly.	
[57]	C-DPOP		Pseudo-tree	✓	•		
[58]	Mailler et al.			✗	Exp.		
[59]	AGAG-ng		Constraint	✓			
[60]	DLB-SDPOP		Pseudo-tree	✓	Exp.	Linear	Exp.
[61]	DEEDS/R-DEEDS		Pseudo-tree	✓	Exp.	Linear	Exp.
[62]	Lezama et al.			✗	Exp.	Linear	
[63]	K1/K2	✓	Constraint	✗			
[64]	Dalapati et al.		Constraint	✓			
[65]	iCO2		Pseudo-tree	✓			
[66]	DisSP		Pseudo-tree	✗			
[67]	DC-DCOP		Constraint	✓	Exp.	Exp.	
[68]	Dis FMC_TA		Constraint	✗	Poly.		
[69]	DPR		Pseudo-tree	Pareto	Exp.	Linear	Exp.
[70]	DynADOPT		Pseudo-tree	✓	Exp.	Exp.	Linear
[71]	DMO-DCOP(DMOBB)	✓	Pseudo-tree	Pareto	Exp.		Exp.
[72]	S-DCOP(SSBDO)	✓	Constraint	✗	Exp.		
[73]	DynCOAA		Constraint	✗			
[74]	BFMS		Factor/Tree	✗	Exp.	Poly.	Linear
[75]	ASR		Constraint	Pareto	Exp.		

## Data Availability

Not applicable.

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
