# Peer review of "Applicability of Multi-Agent Systems and Constrained Reasoning for Sensor-Based Distributed Scenarios: A Systematic Mapping Study on Dynamic DCOPs"

_sensors, 2021, doi:10.3390/s21113807_

Round 1

Reviewer 1 Report

The paper is interesting and the methodology followed in fine. However, several points should be improved:

1) The authors should proofread the paper. There are important English mistakes and sometimes the paper is difficult to follow.

2) The authors should indicate clearly why their study is necessary so it is a step forward with respect previous work. In Section 2.3, after the literature review the authors should indicate the contributions and differences of their work with respect current works.

3) My main point for the authors is the inclusion of a new question (Section 3) related to the methodologies and techniques used to solve the problems DCSP and DCOP. For instance, after reading the definition of DCSP it is very similar to Markov Decision Process used in Reinforcement Learning (RL). Many approaches have been proposed lately for RL. Therefore, I think that they author should indentify and discuss the techniques used, stating advantages and disadvantages in each case.  

Author Response

We deeply appreciate your review and comments. Accordingly, we applied the changes based on your suggestions to the extent of our resources. First, we proofread the paper and fixed the language issues.

We also clarified in section 2.3 the contributions and differences with respect to previous work. In a nutshell, currently there are no systematic reviews of DCOPs, nor a profound search and classification of the corpus for any DCOP family. Said that, we aimed to develop the first systematic mapping on the Dynamic DCOP family, as the subgroup of DCOPs that has been extensively used to model real-world distributed problems with sensors.

Thanks to your comment we have understood that we have not explained this end clear enough. We have improved the redaction of section 4.4 to make it clearer that DCSPs and DCOPs are problems that distributed agents solve in many different ways. Basically, they are divided into search-based, inference-based, and heuristic-based algorithms. Search-based algorithms use search techniques to explore the space of possible solutions, mainly based on best-first and depth-first search. Inference-based algorithms are derived from dynamic programming to solve mathematical optimization problems by breaking it down into simpler subproblems. Heuristic-based are incomplete approaches focused on fast approximations. While ML techniques such as distributed Q-learning and Markov Decision Processes have been an option used, they have only been found in a discrete number of papers and are a small subset of the search-based papers [42,54,62,64].

As we explain in the introduction section, the purpose of our work is to map the literature on dynamic DCOP-based work and which of these are applied in sensor scenarios. Your proposal is very interesting, and we have included it as a future work we would like to explore. We think both approaches are complementary as in your proposal, the question requires a very intensive analysis of each algorithm over a very heterogeneous literature whose resolution requires a very deep expertise in a wide range of techniques and knowledge. In our case, on the contrary, the research questions are focused on general aspects of MAS and DCOP algorithms, such as their complexity, completeness, structure, and properties, leaving out of the scope of the study more technical questions that are usually answered in other types of literature reviews and surveys rather than systematic mappings. In any case, we have clarified these points for a better understanding.

Unfortunately, the inclusion of a new question is not a simple issue, since it implies repeating the whole process associated with the systematic mapping and study presented in the paper, something that is unfeasible to conduct in a reduced time window.

In any case, we have clarified these points for a better understanding.

Reviewer 2 Report

The paper presents a review of distributed constraint optimization  problems (DCOP) approached with the help of multi-agent systems paradigm, with a special focus on sensors and an emphasis on dynamic DCOP. Authors try to do a systematic mapping of the literature, by extensively searching several bibliographical databases and investigating a subset of the articles returned by the search. 

Reading the paper was an interesting experience as I received information about several key issues that a researcher needs to consider when designing a novel MAS for a specific problem. But, with regard to the scientific value of the paper contribution, I found it as being minor.

To bring a major contribution to the literature, the paper needs to clearly differentiate from the review of Fioretto et al, JAIR 2018 (reference 28 of the paper under review). Below, I would bring in several issues that I considered when taking my decision:

  • The topic of DCOP and / or multiagent systems is a huge one. Given the definition of the agents (highlighted in figure 1 of the paper), they are ‘sensing’ the environment in order to collect information. Thus, everything that brings novel information to an agent could be considered as a sensor. The authors should specifically point towards what (specific) sensors do they have in mind, in order to reduce or to better clarify the coverage (scope) of the paper.
  • I do not mind how the authors collected the relevant articles to review; this could be done with the systematic mapping approach or with another methods, but, I would expect that the set of relevant literature to be complete and representative. I have doubts about the search string considered (see table1) and on the selection process highlighted in figure 5, as the total number of publications retrieved is very small. To end up with a total of 40 papers is too little. Either the scope of article under review presented in the introduction was not correctly delimited, being set too large – opposite with what the authors really had in their mind, or the search string was too restrictive. Btw, searching myself only on Scopus with the same query as indicated in table 1, I got a response of more than 2000 entries. The keyword ‘algorithm’ does not help too much, as majority of papers dealing with DCOP are expected to develop or analyse some algorithms.
  • Distributed COP and MAS are inherently linked, as noticed in section 2.2 of the paper. But, agents in a MAS could compete or cooperate for achieving the design goal of the MAS. It is generally agreed that designing MAS with competing agents is much more difficult than with cooperating agents, because agents are selfish, and mechanism design principles (from game theory) should be considered. Dynamicity characteristic of the real problems could lead to MAS approaches with competing agents. Neither in this paper do I see a discussion about competing agents, although there is plenty of literature on this topic. Furthermore, game theoretical principles are widely used together with DCOP for the resolution of MAS. This aspect is essential when designing a MAS, agents could be owned by different stakeholders, environments could be regulated by norms or rules that impose (external) restrictions to the systems, thus, they have a major impact on the definition of the COP. (this comment represents the major issue of the paper with regard to its content).
  • Dynamic DCOP are approached similar with reference [28]. Having D-DCOP put on the title of the paper and on the introduction, I would expect that this review to focus only on D-DCOP, and the dynamic aspect of every problem, issue or feature to be highlighted, in opposition with the (classical) DCOP. RQ1-RQ4 are defined in relation with the dynamic keyword. But, the majority of the analysis of section 4 (Results) barely specifically highlight the dynamic aspect of DCOP. As a reader, it is hard to see within the results section the distinction between dynamic and non-dynamic DCOP.
  • Figure 6 mentions some topics where MAS (with sensors) are applied. But, a topic with just 1 paper is relevant for the discussion? Do there exists other topics which were omitted? (like for example the distributed supply chain management) – probably because of the small set of retrieved references. Better defining what is a sensor-based distributed scenario would help to better focus the paper.
  • From the roadmap of figure 9 I understand that a special emphasis is put on 3 specific dynamic DCOP: multi-objective, probabilistic and proactive. But, discussion and presentation of those specific sorts of D-DCOP is less than half-page (on page 17).
  • The paper has the length of 22 pages. Results and discussion covers only 10 pages. I would extend the coverage of the ‘meat’ of the paper.

As a minor comment, I would point that Scopus covers both IEEE and ACM,  thus searching specifically on IEEE or ACM would bring in many duplicated results.

Author Response

First, we appreciate your comments and we have performed further checks and applied changes according to your suggestions and concerns to the extent of our efforts. Thanks to your comments we understood that we did not explain some ends clear enough. Thus, we have reformulated certain sections accordingly.

We clarify in section 2.3 the contributions and differences with respect to previous work. Fioretto and Leite papers did not conducted systematic reviews of DCOPs, nor a profound search and classification of the corpus for any DCOP family. Hence our contribution is to develop the first systematic mapping on the Dynamic DCOP family, as the subgroup of DCOPs that has been extensively used to model real-world distributed problems with sensors.

The aim of this work is to identify the papers in which the scenario is related to distributed and decentralized sensors to identify which algorithms based on Dynamic DCOPs are deployed on them. The strength of our work is that our systematic mapping does not discriminate against any specific sensor as long as it is part of a distributed sensor network (Distributed Sensor Networks, A Multiagent Perspective, Lesser et al., 2003).

Regarding the obtained corpus, we conducted an SMS process as a well-known systematic procedure to answer the presented research questions to map the literature that converges Dynamic DCOPs as an extended mathematical framework with real-world dynamic scenarios with distributed sensors. As defined in the reference paper of Petersen for conducting SMS (Petersen, 2015), our SMS process has been extensively explained in the paper. Additionally, we also have revised the literature to compare our work with other published SMS, and we have confirmed that our paper includes all the information (or even more) that the one that is typically provided in other SMS in different fields with the aim of making it reproducible and repeatable. For this reason, SMSs tend to be extensive in their methodology and quality assessment. That means that the results of a SMS are as important as the definition of the method followed and rules applied, in order to guarantee its validity and credibility.

Thanks to your comments, we have detected that we have not clearly stated how we defined our queries. We now have specified in the section 5.2 that the queries have been defined and selected through an iterative process that has allowed us to incrementally refine them in order to answer our research questions. The final queries were obtained when the successive iterations comprised a sufficiently high acceptance rate that implied both a minimal modification in the papers obtained and the same papers accepted after applying the inclusion/exclusion criteria. In this way, we tried to reduce possible biases in the definition of our queries. While it is true that the word "algorithm" can be redundant, this has allowed us to eliminate early studies with the same acronym of DCOP but different definitions (Civil Engineering, Medicine, Energy, etc.), as well as papers that superficially mention the topic but do not provide any proposal, without affecting the accepted papers. Moreover, the inclusion of meta-search engines such as Scopus and Web of Knowledge, as well as the snowballing sampling allows us to obtain papers that might be outside those already found in ACM or IEEE and to reinforce the inclusion of potential studies that might have been missed during the search. In any case, and in response to your concerns, we have repeated the queries in this review and have now obtained, with the loosest restrictions, a number quite close to those documented in the study at the time it was carried out (for example, in Scopus it was around 450 papers).

Concerning agents’ behaviour, while it is true that MAS can be composed of both competitive and cooperative agents, in the Dynamic DCOP model, agents are fully cooperative and deterministic by definition (Fioretto et al. 2018). Features found during our study such as privacy cater to more "selfish" behaviours of the agents, but they are still cooperative. The DCOP family that includes adversarial agents are Quantified DCOP (Q-DCOP), introduced by Matsui et al. (A quantified distributed constraint optimization problem, 2010) and Baba et al. (Cooperative problem solving against adversary: Quantified distributed constraint satisfaction problem, 2010). To clarify your concerns, we completed section 2.3 with this explanation.

We agree with the reviewer that the term “Dynamics” may be ambiguous, and then we have explicit specified in the paper in section 2.4, 4.4, and 5.2 what we mean: in our paper, dynamics refer to scenarios that change over time based on the addition/removal of goals, agents, and/or constraints, not on the change of agents' behaviour, or other heterogeneous definitions from the MAS field. In other words, D-DCOPs are represented as a succession of classical DCOPs that must be solved consecutively in a specific time window. From this perspective, our study is focused only on D-DCOPs and our results aim to show which are the properties, characteristics and evolution based on this paradigm. Accordingly, these algorithms have many similarities and are compared in the literature with their predecessors that only solve the problem once.

Regarding figure 6, thanks to the reviewer comments, we understood that it needs further explanation as it may lead to confusion. Certainly, we have found clusters of DCOP papers in different topics such as autonomous vehicles, smart homes, and social simulation, but only one in each field proposes an algorithm based on Dynamic DCOP modelled as described above, and then only one in each field is aligned to our research questions. We introduced some explanation regarding this end to avoid confusions in section 4.1.1.

Regarding Figure 9, we would like to clarify that the mentioned DCOP subgroups emerge from the combination of the D-DCOP with other DCOP families (except for the Proactive ones, which are an extension of the D-DCOP). The properties that emerge from these new subgroups are listed and described in section 4.3. In any case, we have extended the section for better understanding.

Reviewer 3 Report

This clear and interesting paper explores the application of agents in changing environments with distributed and decentralised sensors (e.g., a bunch of drone performing some task falls within this context). In this framework, agents' coordination problem can be seen as a Dynamic Constraint satisfaction Problem (DCOP). The authors propose a review  the literature, creating a systematic landscape that not yet existed. They have examined the scenarios where most proposals have been developed, and the proposal that have arisen to improve the performance, to enhance scalability and to allow designers to integrate off-the-shelf solutions in their projects. Overall, it is a good paper, the English however should be re-read because there are several erroneous words and sentences.

Author Response

We deeply appreciate your review and comments. Accordingly, we applied the changes according to your suggestions, so we proofread the paper and fixed the language issues.

Round 2

Reviewer 1 Report

The paper has been revised according to my comments.

Author Response

We deeply appreciate the feedback provided.

Reviewer 2 Report

I've put extensive comments in the first round, one of them being that I found many more papers when launching in Scopus the query used by the authors. The authors responded very quickly, through I  hardly believe they incorporated novel information in their paper. Thus, I do not think that I could change my decision only by extensively reading their comments to the reviewers.

Also, I do not agree with them that competitive agents paradigm is not in the scope of their paper. Agents in competitive scenario could locally solve a DCOP (including a dynamic one) to solve their specific decision optimization problem. See  for example the Distributed Supply Chain formation problem which is very relevant to their scenario, which fits to the competitive-agents scenario and is no-where mentioned in their paper.  Also, the P2P systems scenario (very frequently approached in the area of distributed systems) could be seen as an instantiation of a competitive-agents scenario. Neither this is mentioned in the review paper and huge literature is available (for both problems mentioned above). 

Author Response

Dear reviewer, we appreciate the feedback provided.

First of all, we understand your concerns and we want to emphasise that we took the review provided seriously and were very thorough in, checking and correcting the suggested changes. While we tried to implement changes in line with your concerns, we were constrained by the viaibility of some requests, due to the scope of our study. In addition, several of the suggested changes are not feasible in such a limited response time (10 days) without detriment to the quality of the process and, consequently, of the study.

On the question of the "competitive" agent scenario, we insist that we have adhered to the definition provided by the literature. While it is stated that the agent teamwork in DCOP problems are "cooperative" for simplicity and by default, we have not deliberately excluded those proposals that present competitive agents (defined as Quantified DCOP), as long as they correspond to the scope of the Dynamic DCOP family (as a sequence of canonical DCOPs).

In reference to the Supply Chain Formation problem you mention, while it does fit with a competitive scenario, the literature mentioned is not related to the dynamic DCOP family. Going deeper, the different proposals by Gaudreault et al. and Penya-Alba et al. use static/single instances of canonical DCOP algorithms, while the proposal by Chli and Winsper, although it poses a dynamic scenario, its solution is a transformation of the Max-Sum algorithm (in its canonical version) to Loopy Belief Propagation, which define the problem outside the DCOP framework model. In reference to P2P systems, although they include competitive scenarios, we have not found references to papers that address this problem by proposing an algorithm of the dynamic DCOP family.

In any case, we have clarified this at the beginning of the paper in the introduction section.